# Kosterlitz-Thouless melting of magnetic order in the triangular quantum Ising material TmMgGaO$_4$

Han Li[1], Yuan Da Liao[2,3], Bin-Bin Chen [1,4], Xu-Tao Zeng[1], Xian-Lei Sheng [1], Yang Qi [5,6✉], Zi Yang Meng [2,7,8✉] & Wei Li [1,9✉]

Frustrated magnets hold the promise of material realizations of exotic phases of quantum matter, but direct comparisons of unbiased model calculations with experimental measurements remain very challenging. Here we design and implement a protocol of employing many-body computation methodologies for accurate model calculations—of both equilibrium and dynamical properties—for a frustrated rare-earth magnet TmMgGaO$_4$ (TMGO), which explains the corresponding experimental findings. Our results confirm TMGO is an ideal realization of triangular-lattice Ising model with an intrinsic transverse field. The magnetic order of TMGO is predicted to melt through two successive Kosterlitz–Thouless (KT) phase transitions, with a floating KT phase in between. The dynamical spectra calculated suggest remnant images of a vanishing magnetic stripe order that represent vortex–antivortex pairs, resembling rotons in a superfluid helium film. TMGO therefore constitutes a rare quantum magnet for realizing KT physics, and we further propose experimental detection of its intriguing properties.

[1] Key Laboratory of Micro-Nano Measurement-Manipulation and Physics (Ministry of Education), School of Physics, Beihang University, Beijing 100191, China. [2] Beijing National Laboratory for Condensed Matter Physics and Institute of Physics, Chinese Academy of Sciences, Beijing 100190, China. [3] School of Physical Sciences, University of Chinese Academy of Sciences, Beijing 100190, China. [4] Arnold Sommerfeld Center for Theoretical Physics, Center for NanoScience, and Munich Center for Quantum Science and Technology, Ludwig-Maximilians-Universität München, Fakultät für Physik, D-80333 München, Germany. [5] Center for Field Theory and Particle Physics, Department of Physics and State Key Laboratory of Surface Physics, Fudan University, Shanghai 200433, China. [6] Collaborative Innovation Center of Advanced Microstructures, Nanjing 210093, China. [7] Department of Physics and HKU-UCAS Joint Institute of Theoretical and Computational Physics, The University of Hong Kong, Pokfulam Road, Hong Kong, China. [8] Songshan Lake Materials Laboratory, Dongguan, Guangdong 523808, China. [9] International Research Institute of Multidisciplinary Science, Beihang University, Beijing 100191, China. ✉email: qiyang@fudan.edu.cn; zymeng@hku.hk; w.li@buaa.edu.cn

Kosterlitz–Thouless (KT) physics bestows interesting mechanism of phase transition upon two-dimension (2D) interacting system with a continuous symmetry. Although such symmetry is not allowed to break spontaneously at any finite temperature[1], phase transition can still take place from the high-temperature disordered phase to a KT phase with quasi-long-range order, which has a topological root in the binding of the vortex and antivortex pair[2,3]. Experimentally, the KT transition has been observed in thin helium films[4] and ultracold 2D Bose gases[5,6], etc. Two distinct types of elementary excitations, i.e., phonons and rotons, play essential roles in the related superfluid phenomena[7,8], and they are important for understanding liquid helium thermodynamics[9]. Besides interacting bosons in liquid and gas, there are also theoretical proposals of KT transitions in solid-state magnetic systems such as the 2D classical XY[2,3] and the frustrated quantum Ising models[10]. However, to date, the material realization of the KT transition in 2D magnets has rarely been reported.

In the mean time, the search of exotic quantum magnetic states in the triangular-lattice spin systems—the motif of frustrated magnets—has attracted great attention over the decades. Experimentally, the triangular-lattice quantum magnets have been synthesized only very recently, including compounds $Ba_3CoSb_2O_9$[11–14], $Ba_8CoNb_6O_{24}$[15,16], and a rare-earth oxide $YbMgGaO_4$—which has been suggested as a quantum spin liquid candidate[17–20]—whereas an alternative scenario of glassy and disorder-induced state has also been proposed recently[21–23]. On the other hand, an Ising-type triangular antiferromagnet $TmMgGaO_4$ (TMGO, with $Yb^{3+}$ replaced by another rare-earth ion $Tm^{3+}$)[24–26], as shown in Fig. 1a and explained in details in this study, is the successful material realization of a quantum magnet with strong Ising anisotropy.

In this work, we construct the microscopic model of TMGO and employ two state-of-the-art quantum many-body simulation approaches: the exponential tensor renormalization group (XTRG)[27] and quantum Monte Carlo (QMC) equipped with stochastic analytic continuation (QMC-SAC)[28–32], to calculate both the thermodynamic and dynamic properties. By scanning various parameters and fit our simulation results to the existing experimental data[24–26], we find TMGO realizes a triangular-lattice transverse-field Ising model and determine accurately its model parameters. Based on this, we conclude that TMGO should host the celebrated KT phase and further predict several prominent features to be observed in TMGO, inspired by the experimental measurements for detecting KT physics in a superfluid thin film[4]. It is worthwhile to point out that our calculation of quantum fluctuations goes beyond the linear spin-wave approximation in ref. [26] and puts the system in the clock-ordered (later melted through KT transitions), rather than disordered, regime. Therefore, our methods and results do not only explain the experimental findings but, more importantly, establish a protocol for acquiring equilibrium and dynamic experiments of strongly correlated quantum materials, such as TMGO, in an unbiased manner.

## Results

**Microscopic spin model.** Due to strong spin–orbit coupling and crystal electric field splitting, TMGO can be described as an effective spin-1/2 model with strong easy-axis anisotropy, i.e., a triangular-lattice Ising model (TLI), as shown in Fig. 1a. First-principle calculations of the TMGO material are performed based on the density functional theory (DFT)[33–36] where we see a large easy-axis anisotropy of room-temperature energy scale. By

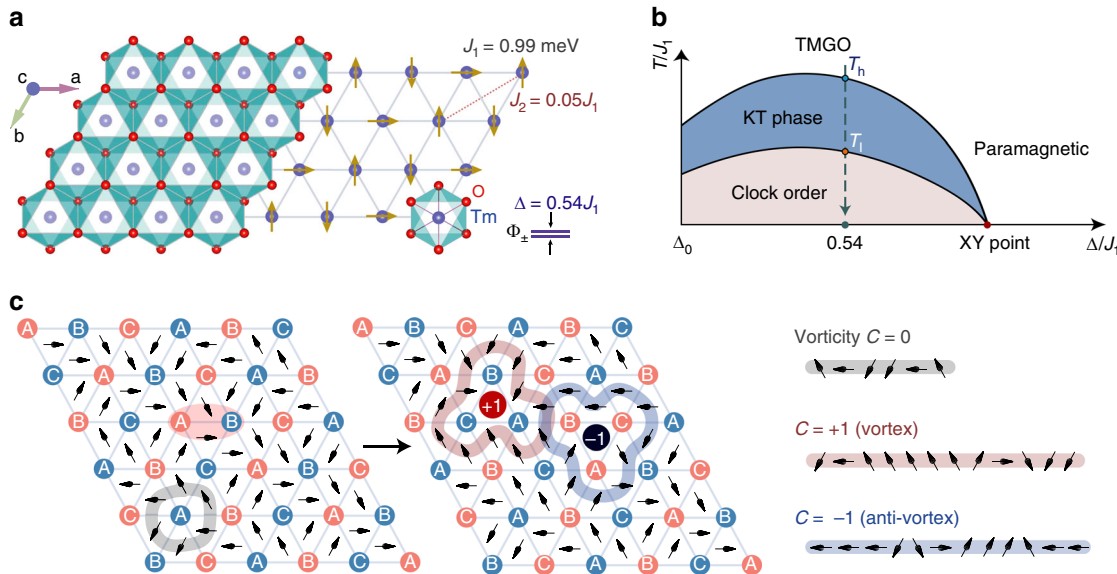

**Fig. 1 The crystal structure, phase diagram, and spin texture in quantum magnet TMGO. a** The $Tm^{3+}$ ions, with an energy splitting $\Delta$ between two lowest non-Kramers levels $|\Phi_{\pm}\rangle$ constitute an effective spin-1/2 model on the triangular lattice, with $J_1$ and $J_2$ interactions. An illustration of spin structure in the clock phase is provided, where the spin-up and spin-down arrows are along the magnetic easy c-axis and the horizontal arrow stands for superposition of spin up and down, i.e., $|\rightarrow\rangle$. **b** The schematic phase diagram of quantum TLI model, there exists a quantum critical point on the horizontal axis with emergent spin XY (i.e., U(1)) symmetry, which extends into an intriguing KT phase at finite $T$. $\Delta > \Delta_0$ stands for transverse fields where the clock order is stabilized in the ground state, in the $J_1$–$J_2$ model with small $J_2$ coupling. The vertical arrowed line along $\Delta = 0.54J_1$ represents the TMGO material, with two KT transitions at $T_h$ and $T_l$, respectively. **c** The magnetic stripe order, with the red sites for spin up ($m^z = 1/2$) and blue ones spin down ($m^z = -1/2$) on three sublattices A, B, and C. The pseudo spins, i.e., complex order parameters $\psi$ in Eq. (2), are plotted as arrows rotating within the plane, and a vortex-antivortex pair is created by flipping simultaneously two spins within the red oval in the left subpanel. As tracked along the paths (exemplary paths are indicated in the plot), topological charge $C = \pm 1$, corresponding to $2\pi$ clockwise/counterclockwise angle, emerges when the pseudo spins wind clockwise around the vortex/antivortex, and zero vorticity appears when counting the pseudo-spin winding of both (or no defect at all).

comparing the DFT energies of antiferromagnetic spin configuration with a ferromagnetic one, one finds the former has a lower energy and the coupling strength can be estimated as a few tenths of meV. Density distribution of $4f$ electrons in TMGO can also be obtained, where it is observed that the $4f$ electrons of $Tm^{3+}$ are coupled via superexchange mediated by $2p$ electrons of $O^{2-}$ within the triangular-lattice plane (Supplementary Note 1).

In ref. [25], the authors took the lowest two levels $|\Phi_{\pm}\rangle$ of $Tm^{3+}$ as non-Kramers doublet and construct a classical TLI with both nearest-neighbor (NN) and next-nearest-neighbor (NNN) interactions to account for the absence of zero-point entropy observed in experiments. Substantial randomness was also introduced to explain the smooth magnetization curves even at a very low temperature. Later on, inelastic neutron scattering (INS) results of TMGO reveal a clear magnon band[26], suggesting the influence of coupling randomness should be modest in TMGO and an adequate modeling of the material shall include in the Hamiltonian non-commuting terms with quantum fluctuations. As the Kramers theorem is absent in $Tm^{3+}$ system with total angular momentum $J = 6$, a small level splitting $\Delta$ between the quasi-doublet $|\Phi_{\pm}\rangle$ is involved, as shown in Fig. 1a. Therefore, a quantum TLI model was proposed[26], with spin-1/2 Hamiltonian

$$H_{TLI} = J_1 \sum_{\langle i,j \rangle} S_i^z S_j^z + J_2 \sum_{\langle\langle i,j \rangle\rangle} S_i^z S_j^z - \sum_i (\Delta S_i^x + h\, g_{\parallel} \mu_B\, S_i^z), \quad (1)$$

where $\langle , \rangle (\langle\langle , \rangle\rangle)$ stands for NN (NNN) couplings $J_1$ ($J_2$), $\Delta$ the energy splitting between $|\Phi_{\pm}\rangle$ (i.e., the intrinsic transverse field), and $h$ is the external magnetic field. $g_{\parallel} = 2J g_J$ constitutes the effective spin-1/2 $g$ factor, with $g_J$ the Landé factor.

The phase diagram of quantum TLI has been studied intensively with analytic and numeric methods in the past[10,37–39], and is schematically shown in Fig. 1b. We indicate the TMGO model parameter with the vertical arrow (the determination of parameters is given below). From high to low temperatures, the system first goes through an upper KT transition at $T_h$ from the paramagnetic phase to a KT phase with power-law (algebraic) spin correlations. At a lower temperature $T_l$, the system enters the clock phase with a true long-range order depicted in Fig. 1a. This three-sublattice clock order breaks the discrete lattice point group and the $Z_2$ spin symmetries, giving rise to a low but finite transition temperature $T_l$.

Increasing the next-nearest-neighbor coupling $J_2$, say, at $J_2/J_1 = 0.2$, we find the static magnetic structure factor develops a stripe order[40] with structure factor peak at M point (see, e.g., Fig. 4a) of the Brillouin zone (BZ) (see Supplementary Notes 2 and 3). This magnetic stripe order, as shown in Fig. 1c, has been observed previously in TLI material $AgNiO_2$, where $J_2$ coupling is relatively strong (~$0.15 J_1$, along with other interactions) and the exotic KT physics is absent there[41]. In TMGO, however, the clock order wins over the stripe order as $J_2/J_1 \simeq 0.05$ is relatively small in this material. Nevertheless, as will be shown below, a ghost of the stripe order—the M rotonlike modes—remains in the spin spectrum[26], which turns out to be related to a vortex–antivortex pair excitation in the topological language (see Discussion section).

**Thermodynamics and parameter fittings.** The model parameters in Eq. (1) can be accurately determined through fitting the available experimental data of TMGO[24–26], from which we find $J_1 = 0.99$ meV, $J_2 = 0.05 J_1$, $\Delta = 0.54 J_1$ and $g_{\parallel} = 13.212$. We present in Fig. 2 the calculated thermodynamic quantities and their experimental counterparts[25,26], where excellent agreements are seen. In Fig. 2a, at $T > 30$ K, the magnetic entropy $S_m$ approaches $R \ln 2$, corresponding to the high-temperature paramagnetic phase with effective spin-1/2. As temperature decreases, $S_m$ gradually

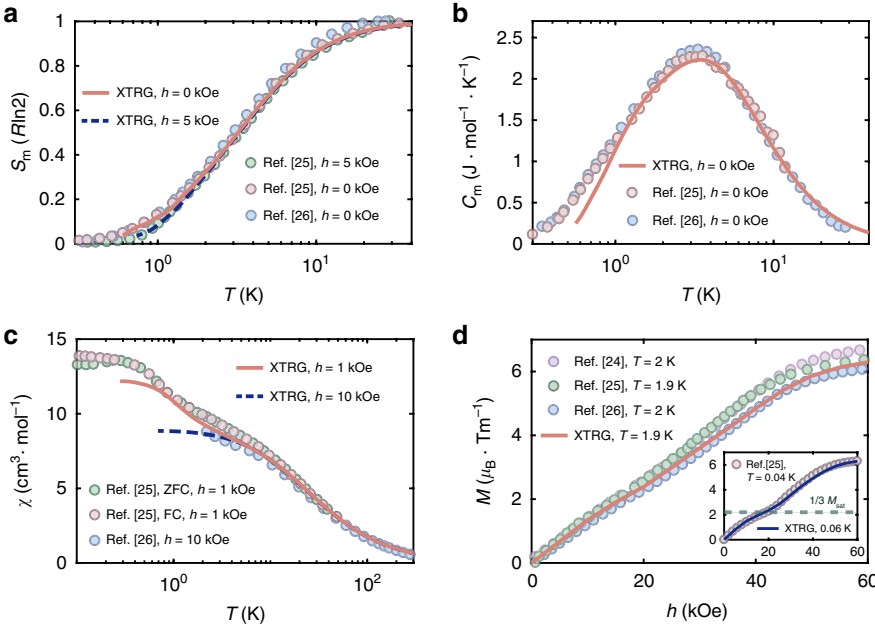

**Fig. 2 Thermodynamic measurements and XTRG fittings to experimental results.** Agreements between the experimental curves (taken from various independent measurements[24–26]) and numerical results can be seen in all panels, down to very low temperatures. **a** Includes two experimental entropy curves under fields $h = 0$ and 5 kOe[25,26], and the specific heat data shown in **b** are under zero field. The magnetic susceptibility $\chi$ is shown in **c**, which follows first the Curie-Weiss law at high temperature, i.e., $\chi \sim C/(T - \Theta)$ with $\Theta \simeq -19$ K (see Supplementary Note 5) and then exhibits at ~10 K a shoulder structure, signifying the onset of antiferromagnetic correlation. For $T \lesssim T_h^* \sim 4$ K, $\chi$ rises up again and eventually converges to a finite value as $T$ decreases to below $T_l^* \sim 1$ K. These anomalous susceptibility curves can be fitted very well by our simulations and naturally understood within the TLI model. For the magnetization curves in **d**, the perfect consistency between numerics and experiments hold for both intermediate-$T$ ($\simeq 2$ K) and the low-$T$ (40–60 mK) curves. The latter is shown in the inset, where a quasi-plateau at $M \simeq \frac{1}{3} M_{sat}$ becomes prominent, with $M_{sat} = J g_J \mu_B$ the saturation magnetization of TMGO.

releases throughout the intermediate-temperature regime and approaches zero below $T_1^* \simeq 1$ K, as the long-range clock order develops. In Fig. 2b, the very good agreement in magnetic-specific heat $C_m$ extends from high $T$ (~30 K) all the way down to low temperature $T \sim 1$ K.

In the fittings of the magnetic entropy and specific heat curves in Fig. 2a, b, we rescale the $T$-axes (in the unit of $J_1$) to lay the model calculations on top of the experimental measurements, and in this way we find the optimized $J_1 = 0.99$ meV. In both plots, the $y$-axes scaling ratios are associated with the ideal gas constant $R = 8.313$ J mol$^{-1}$ K$^{-1}$ and thus fixed. In Fig. 2c, we fit the magnetic susceptibility $\chi(h) = \frac{M(h)}{h}$, with $M(h)$ the uniform magnetization (per Tm$^{3+}$), under external fields of a small value $h = 1$ kOe and a larger one 10 kOe. As shown in Fig. 2c, by setting the effective Landé factor $g_\parallel = 13.212$, we fit both susceptibility curves very well. This completes the model parameters in the Hamiltonian Eq. (1) (see more fitting details in Supplementary Note 4) and note this parameter set also leads to accurate entropy results at 5 kOe when put in a direct comparison with the experimental data in Fig. 2a.

With the parameters $J_1$, $J_2$, $\Delta$, and $g_\parallel$ determined from the above fittings, we can compute the magnetization curve $M(h)$ and compare it directly with several independently measured experimental curves in Fig. 2d. It is noteworthy that there exists a turning point at about 1/3 magnetization, under a magnetic field around 20–25 kOe. It becomes clearer as $T$ decreases further down to 40 mK (see the inset of Fig. 2d), suggesting the existence of field-induced quantum-phase transition in TMGO. This sharp change of behaviors can also be witnessed in the specific heat curves under various magnetic fields (see Supplementary Note 6), which can also be understood very well within the set of parameters obtained above.

Lastly, we briefly discuss the scalings in the uniform susceptibility $\chi$ in Figs. 2c and 3, where susceptibility curves under more external fields are computed theoretically in the latter plot as a complementary. These $\chi$ data reflect the two-step establishment of magnetic order as $T$ lowers. As pointed out in refs. [42,43], a universal

scaling $\chi(h) = h^{-\alpha}$ appears for small fields $h$ with $\alpha = \frac{4 - 18\eta(T)}{4 - 9\eta(T)}$ in the KT regime, where $\eta(T) \in [\frac{1}{9}, \frac{2}{9}]$ is the anomalous dimension exponent of the emergent XY-order parameter varying with temperature. For $T = T_l$, $\eta = \frac{1}{9}$ and $\chi(h) \sim h^{-2/3}$, which diverges as $h$ approaches zero, whereas above some higher temperature $T_h^*$, $\eta = \frac{2}{9}$ and $\chi(h)$ remains a constant vs. $h$. Therefore, at small external field, the increase of $\chi$ at intermediate $T$ reflects the decrease of $\eta(T)$ vs. $T$ and such enhancement becomes less prominent for a relatively larger field, say, $h = 10$ kOe. This salient difference indeed can be noticed in the experimental and our numerical curves in Fig. 2c. Moreover, in Fig. 3 we plot in the inset susceptibility curves under various magnetic fields $h$ between 6.5 kOe and 9 kOe, where the power-law scaling is shown explicitly and the anomalous exponent $\eta$ can be extracted therein.

**Spin spectra and magnetic structure factors.** Spin frustration can lead to strong renormalization effects, which in turn gives rise to interesting spectrum in dynamics. Here we employ the QMC-SAC approach[28–32] to compute the spin spectra $S(\mathbf{q}, \omega)$ from $S^z$ spin correlations, at various temperatures (see Methods). The obtained spectra, with model parameters determined from equilibrium data fittings, are plotted in Fig. 4 and are compared directly with the INS results[26]. Figure 4a depicts the spin spectrum inside the clock phase, at a low temperature $T = 0.5$ K. As the clock phase is of discrete symmetry breaking, the $S(\mathbf{q}, \omega = 0)$ at K point signals the Bragg peak of the clock order and there exists a small gap ~0.1 meV between the $\omega = 0$ and finite $\omega$ spectra, consistent with the INS result. The rotonlike modes are also clearly present in the QMC-SAC results, with an energy gap about 0.4 meV, in quantitative agreement with that in ref. [26].

Figure 4b is the QMC-SAC spectrum calculated according to the parameters ($\Delta/J_1 \simeq 1.36$, $J_2/J_1 \simeq 0.046$) given in ref. [26]. As mentioned in the introduction, we find, via spin structure factor calculations, that such set of parameters actually put the model in the disordered paramagnetic phase with $\Delta > \Delta_c \sim 0.8J_1$. It is possible that the fitting scheme adopted in ref. [26] is based on mean-field treatment and cannot capture the quantum fluctuations inherent to the quantum TLI model and the material TMGO. This is a clear sign that the unbiased quantum many-body calculation scheme in our work is the adequate approach to explain the experimental results.

We continue with the correct parameter set and rise the temperature to $T = 1.45$ K in Fig. 4c. It is interesting to see that the dispersion still resembles that in the clock phase of Fig. 4a but with a vanishing gap at the K point and softened M roton modes. To show it more clearly, we plot the intensity at M in Fig. 4d, where the roton gap gets reduced as $T$ increases, with substantially broadened linewidths. As M rotonlike excitations can be related to vortex-pair excitation (see Discussions), this softening of M roton is consistent with the scenario of vortex proliferations near the upper KT transition. Such remarkable spectra constitute a nontrivial prediction to be confirmed in future INS experiments.

Besides, the static magnetic structure factor $S(\mathbf{q}) = \sum_{i,j} e^{i\mathbf{q}\cdot(\mathbf{r}_i - \mathbf{r}_j)} \langle S_i^z S_j^z \rangle$ are also simulated, where $\mathbf{r}_i$ and $\mathbf{r}_j$ run throughout the lattice. Figure 5a shows the temperature dependence of $S_K$ and $S_M$, where one observes an enhancement of $S_M$ at intermediate temperature, signifying its closeness towards the stripe order. At $T < T_1^*$, the enhancement of $S_M$ vanishes and instead the $S_K$ intensity becomes fully dominant. Figure 5b–d show the $S(\mathbf{q})$ results at low ($T = 0.57$ K), intermediate ($T = 2.2$ K), and high temperaure ($T = 4.5$ K). In the clock phase, $S(\mathbf{q})$ evidently peaks at the K point, the ordering wavevector of the three-sublattice clock phase, whereas in the intermediate temperature regime, notably there exists a "ghost" peak at the M point, manifesting the

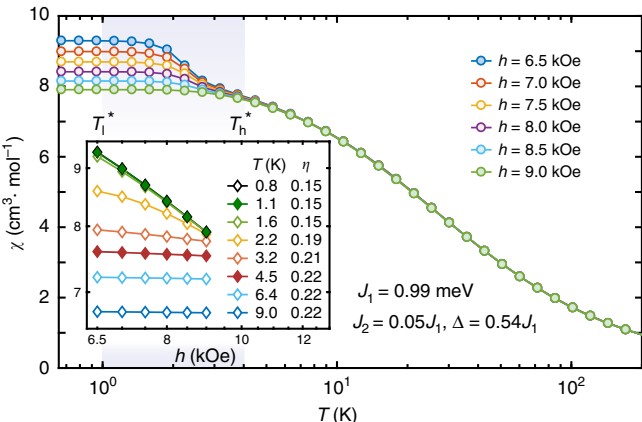

**Fig. 3 Algebraic scaling of uniform magnetic susceptibility.** At high temperature $T \gtrsim 4$ K, $\chi(h, T)$ remains independent of $h$ for small fields, which indicates an exponent of $\alpha = 0$ in $\chi(h, T) \sim h^{-\alpha(T)}$, whereas for $1$ K $\lesssim T \lesssim 4$ K (shaded regime), $\chi(h, T)$ exhibits an universal scaling vs. $h$ with $\alpha \neq 0$. For temperatures below $T_1^* \sim 1$ K, the susceptibility ceases to increase as the system orders into the clock phase. In the inset, $\chi(h, T)$ vs. $h$ curves are presented in the log–log plot, where the algebraic scaling manifests itself in the KT phase (and in the clock phase due to the saturation). The fitted $\eta$ values decrease from 2/9 gradually to ~0.15, the deviation of the latter from the expectation of 1/9 is ascribed to limited lattice size ($L = 12$ torus) adopted in the QMC calculations.

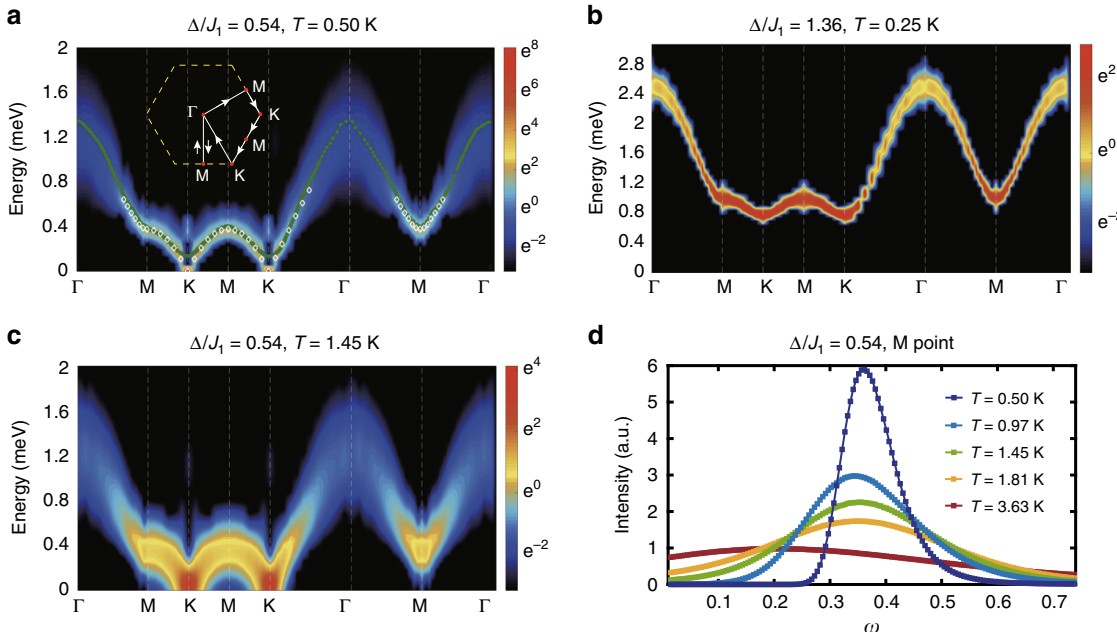

**Fig. 4 QMC-SAC spectra of TMGO at finite temperature. a** Shows the calculated spin spectrum of TLI at $T = 0.5$ K, with model parameters determined in this work and computed on an $L = 36$ torus geometry. The computed spectrum agrees excellently with the experimental INS results taken from ref. [26], whose peak positions are shown as the green dotted line. The white diamonds represent the bottom-part excitation results without resorting to analytical continuation, which agree very well with the QMC-SAC results. The path consists of the Γ-M-K-M-K-Γ loop and a Γ-M-Γ vertical mini-loop, as shown in the inset. **b** Spectrum with the parameter set given in ref. [26], which clearly fails to describe the material. In **c**, we plot in the spin spectra with model parameters in **a** but at a higher temperature 1.45 K. Compared with **a**, the K point gap gets smoothed and the rotonlike gap reduced. We collect the M point intensity vs. $\omega$ and plot in **d**, where the linewidth near the rotonlike minima is substantially broadened as $T$ increases, suggesting strong fluctuations and vortex proliferation in the system.

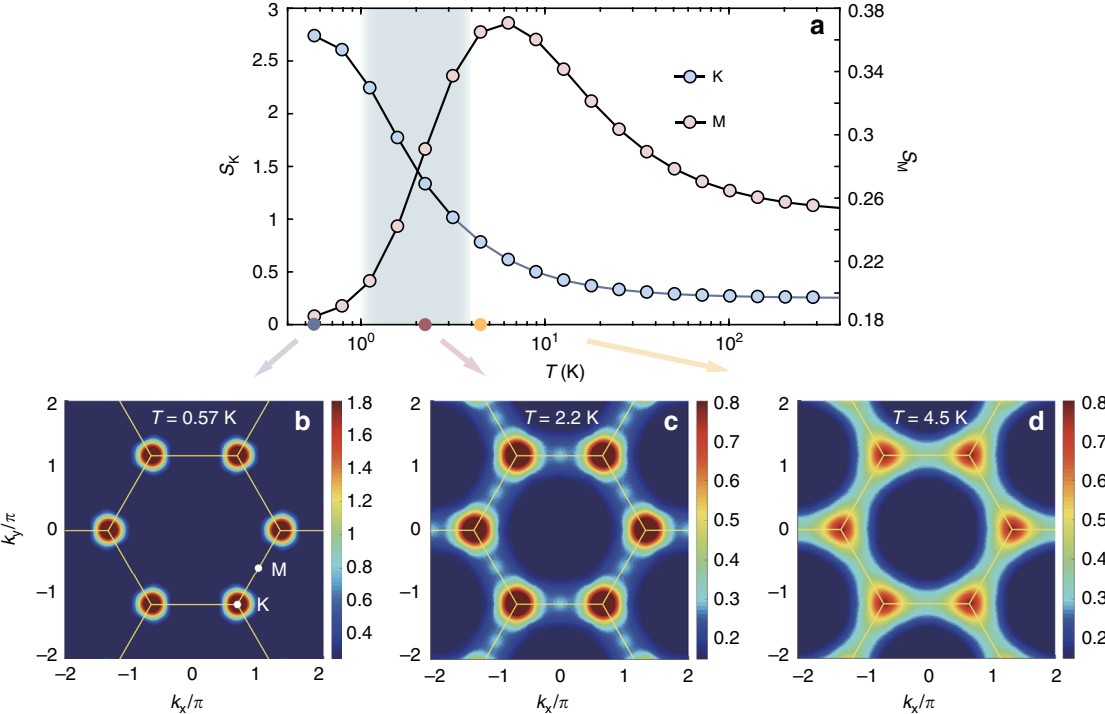

**Fig. 5 Static magnetic structure factor of TMGO at finite temperatures. a** The temperature dependence of the structure factor $S(q)$ at M and K points indicated in **b**. The $S_M$ value enhances anomalously in the intermediate temperatures, whereas the $K$-point peaks become significantly strengthened, representing the full establishment of the three-sublattice clock order at $T = 0.57$ K as shown in **b**. **c** Shows the $S(q)$ contour at $T = 2.2$ K, where a "ghost" peak appears at M, representing short-range stripy correlation. In **d**, as $T$ further enhances to 4.5 K, the K peak clearly weakens and spreads out and the M ghost peak becomes virtually invisible.

existence of short-range stripe order selected by thermal fluctuations. These interesting features are gone at higher $T = 4.5$ K, where strong fluctuations considerably weaken the structure factor peaks.

## Discussion

As postulated by Landau, the phonon–roton spectrum play an essential role in understanding low-$T$ thermodynamics and critical velocities of superfluidity, etc, in liquid helium, where the rotons are believed to be related to local vorticity of the fluid[7]. Roton constitutes a minima in the spectrum at finite momentum and energy, and has been regarded as a quantum analog of hydrodynamic vortex ring, as coined "the ghost of vanished vortex ring" by Onsager[44]. On the other hand, as derived from a trial wavefunction introduced by Feynman[8], the roton excitation has energy $\epsilon_{\mathbf{q}} \sim \hbar \mathbf{q}^2 / [2m \, S(\mathbf{q})]$, with $m$ is the helium atom mass and $S(\mathbf{q})$ the structure factor peak, and it is therefore also associated with an incipient crystalization Bragg peak competing with superfluidity[45]. Rotonlike excitations are also found in thin helium films[46] and frustrated triangular Heisenberg (TLH) magnets. In the latter case, M point rotonlike modes were predicted theoretically[47–50] and confirmed in recent experiments[13,14], whose nature is under ongoing investigations[51,52]. Notably, it has been proposed that the M rotons in TLH can be softened by further enhancing spin frustration (and thus quantum fluctuations)[52] or thermal fluctuations[53], which melts the long-range or incipient semi-classical 120° order, driving the system into liquid-like spin states.

In the frustrated magnet TMGO, as mentioned earlier in Fig. 4, there exists rotonlike modes with large density of states, which becomes softened even at low temperatures, melting the clock order, and strongly influences thermodynamics of the system. Similar to that in liquid helium, roton in TMGO has also a topological origin: we demonstrate below that the rotonlike modes represent bound states of topological vortex excitations, via a pseudo-spin mapping of the spin stripe order. As shown in Fig. 1c, the stripe order constitutes a proximate competing order to the clock state. Although it eventually gets perished in TMGO at low $T$, the stripe order leaves a "ghost image" in the excitation spectrum, i.e., rotonlike dip along the Γ-M-Γ path in Fig. 4a. Correspondingly, there exists an incipient $S_M$ peak in the static structure factor at intermediate $T$ (Fig. 5c), i.e., $\epsilon_{\mathbf{q}} \sim 1/S(\mathbf{q})$, similar to rotons in the superfluid helium discussed above[8,45].

On top of the spin stripe order, the M rotonlike excitation can be related to a locally bounded vortex pair, some form of "rotational motion" happening in TMGO. We perform a pseudo-spin mapping

$$\psi = m_A^z + e^{i2\pi/3} \, m_B^z + e^{i4\pi/3} \, m_C^z, \tag{2}$$

where $\psi = |\psi| e^{i\theta}$ is the complex order parameter[37], i.e., the pseudo-spin. As shown in Fig. 1c, $\psi$ is located in the center of each triangle, with emergent XY degree of freedom $\theta$. In Eq. (2), $m_\gamma^z = \pm 1/2$ represents the spin-up(-down) of corresponding spin $S^z$ components at $\gamma$-sublattice ($\gamma = $ A, B, C). This mapping helps establishing a Landau–Ginzburg theory of TLI[10,37] and the clock order shown in Fig. 1a corresponds to a ferromagnetic order of pseudo spins.

As shown in Fig. 1c, we create a vortex pair by applying $S^x$ operator on two adjacent sites to simultaneously flip their spin orientations. We note that any closed loop enclosing only the vortex defect (red dot in Fig. 1c) leads to a winding number 1 (modulo $2\pi$), whereas those around the antivortex (black dot) to $-1$. Zero winding number can be counted when a pair of defects (or no defects at all) are enclosed by the loop. Moreover, one can further move the vortex on the triangular lattice, such as in a "tight-binding" model, by flipping spins on further neighboring

sites, which naturally leads to a quadratic-type low-energy dispersion near M point along Γ-M-Γ (see Supplementary Note 7). As the ghost peak in Fig. 5c only suggests a short-range stripe correlation, the vortex pair can thus move only within a small cluster with incipient stripe order, i.e., they are bounded. The vortex pair only unbinds at the upper KT transition $T_h$ where vortices are proliferated[2,3], as seen by the "softening" of M rotonlike mode in dynamical spectrum in Fig. 4.

To conclude, in this work, we have established a protocol of understanding and explaining experiments of frustrated magnets in an unbiased manner, with XTRG and QMC-SAC machinery. The thermodynamic and dynamic results of TMGO are captured to great accuracy, thus allowing comprehensive studies of KT physics therein. At intermediate temperature, the KT phase of TMGO realizes a magnetic analog of 2D superfluid phase with several intriguing properties: (i) there emerges a spin XY symmetry and correspondingly complex order parameter $\psi$, which bears quasi-long-range correlation and phase coherence; (ii) the finite-$T$ spin spectrum contains the long-wavelength magnon and competing gapped rotonlike modes near the BZ boundary with energy signifying the binding of vortex–antivortex pair, which plays a key role in determining finite-$T$ phase diagram of the system; (iii) the quasi-long-range XY order melts and the TMGO becomes paramagnetic as $T$ is above $T_h$, driven by the proliferation of vortex excitations, in analogy to the superfluid transition in a helium thin film[4,54]. We note that, different from the liquid helium, TMGO has two-temperature scales that outline the intermediate-temperature KT phase. Similar separation of scales has been seen in other quantum magnetic materials, by temperature[53,55] or spatial dimensions[56], but the KT phase is the first time to be seen. Nevertheless, the $T_l^*$, $T_h^*$ in the present work constitute tentative estimated of two KT transition temperatures roughly from thermodynamics, which still needs to be precisely determined both numerically and experimentally in the future.

The extraction of anomalous exponents $\eta(T)$ constitutes another interesting future study. The exponent $\eta(T)$ of the KT phase appears in the algebraic correlations in $\langle S_i^z S_j^z \rangle \sim |r_i - r_j|^{-\eta(T)}$ and seems rather indirect to measure in solid-state experiments. Nevertheless, as discussed above in Fig. 3, $\eta(T)$ can be determined from the uniform susceptibility $\chi$, a routine magnetic quantity in experiments. Moreover, it would be interesting to check several distinct predictions of KT physics in this 2D magnetic material. One renowned phenomenon is the universal jump in superfluid density at the KT transition[57], as observed experimentally in helium film[4]. Through calculations of the $q$-clock model ($q = 5, 6$), people have revealed universal jumps in the spin stiffness at both upper and lower KT transitions[58], which certainly is an interesting prediction to check in TMGO. Dynamically, nuclear magnetic resonance measurements of relaxation time $1/T_1$ can be conducted, which probe signals of low-energy magnetic dissipations at the KT transition where vortices proliferate. Besides, non-equilibrium thermodynamics such as thermal transport, would also be very worthwhile to explore in the TMGO magnet.

## Methods

**Quantum many-body computations.** In this work, we combined two many-body numerical approaches: QMC[37] and XTRG, the latter method is recently introduced based on matrix product operators (MPOs) and logarithmic temperature scales[27]. XTRG is employed to simulate the TLI down to temperatures $T < 0.5$ K on YC $W \times L$ geometries up to width $W = 9$ with various lengths up to $L = 12$. Both dynamical and equilibrium properties are simulated, with the purpose of fitting the experimental data and obtaining the right parameters, as well as to make predictions for experiments. The QMC is performed in the space-time lattice of $L \times L \times L_\tau$, where $L = 36$ and $L_\tau = \beta/\Delta\tau$ with $\Delta\tau = 0.05$ and $\beta \equiv 1/T$. The space-time configuration is written in the $S_{i,\tau}^z$ basis with both local and Wolff-cluster updates to overcome the long autocorrelation time. As the QMC method is standard, we will only introduce the SAC scheme below and leave the QMC itself to the Supplementary Note 8.

**Exponential thermal tensor network method**. For the calculations of equilibrium properties, we start from a high-$T$ density matrix $\hat{\rho}(\Delta\tau) = e^{-\Delta\tau H}$, whose MPO representation can be obtained conveniently and accurately (up to machine precision), at a small $\Delta\tau \sim 10^{-3\sim-4}$. One way to obtain such an accurate MPO representation is to exploit the series-expansion thermal tensor network technique[59] via the expansion

$$\hat{\rho}(\Delta\tau) = e^{-\Delta\tau\hat{H}} = \sum_{n=0}^{N_c} \frac{(-\Delta\tau)^n}{n!} \hat{H}^n. \tag{3}$$

Given the $\hat{\rho}(\Delta\tau)$ representation, traditionally one evolves $\hat{\rho}(\beta)$ linear in $\beta$ to reach various lower temperatures, i.e., $\beta = L_\tau\Delta\tau$ increases by a small value $\Delta\tau$ after each step by multiplying $\hat{\rho}(\Delta\tau)$ to the density matrix[60]. However, this linear scheme is not optimal in certain aspects, and encounters challenges in generalization to 2D. Instead, recent study shows that the block entanglement entropy of MPO is bound by $S_E \leq a \ln\beta + \text{const.}$ at a conformal critical point, with $a$ an universal coefficient proportional to the central charge[27]. This suggests an exponential procedure of performing cooling procedure. Based on this idea, we have developed the XTRG method, which turns out to be highly efficient in simulating both one-dimensional (1D) critical quantum chains and various 2D lattice systems[27,53,61].

In XTRG, we cool down the system by multiplying the thermal state by itself, i.e., $\rho_0 \equiv \rho(\Delta\tau)$, $\rho_1 \equiv \rho_0 \cdot \rho_0 = \rho(2\Delta\tau)$, thus $\rho_n \equiv \rho_{n-1} \cdot \rho_{n-1} = \rho(2^n\Delta\tau)$, and reach the low-$T$ thermal states exponentially fast. Efficient compression of MPO bonds is then required to maintain the cooling procedure, where a truncation scheme optimizing the free energy and in the mean time maintaining the thermal entanglement, is involved. One advantage of XTRG is the convenience and high efficiency to deal with long-range interactions after the quasi-1D mapping. For the TLI model with NN ($J_1$) and NNN ($J_2$) interactions considered in this work, we map the 2D lattice into a quasi-1D geometry following a snake-like path. The Hamiltonian thus contains "long-range" interactions and has an efficient MPO representation with geometric bond dimension $D_H = 2W + 2$, with $W$ the width of the lattice. In XTRG calculations, the computational costs scale with power $O(D^4)$, with $D$ the retained bond dimension in MPO, which is chosen as large as 500–600 in the present study, assuring accurate thermodynamical results down to sub-Kelvin regime.

**QMC-SAC approach**. We exploit the path integral QMC[37], equipped with SAC approach, to compute the dynamical properties. The time displaced correlated function, defined as $G(\tau) = \langle S^z(\tau)S^z(0)\rangle$, for a set of imaginary times $\tau_i(i = 0, 1, \cdots, L_\tau)$ with statistical errors can be obtained from QMC simulations. By SAC method[28,29,31,32], the corresponding real-frequency spectral function $S(\omega)$ can be obtained via $S(\tau) = \int_{-\infty}^{\infty} d\omega S(\omega)K(\tau,\omega)$, where the kernel $K(\tau,\omega)$ depends on the type of the spectral function, i.e., fermionic or bosonic, finite or zero temperature. The spectra at positive and negative frequencies obey the relation of $S(-\omega) = e^{-\beta\omega}S(\omega)$ and we are restricted at the positive frequencies and the kernel can therefore be written as $K(\tau,\omega) = \frac{1}{\pi}(e^{-\tau\omega} + e^{-(\beta-\tau)\omega})$. To work with a spectral function that is itself normalized to unity, we further modify the kernel and the spectral function, and arrive at the transformation between the imaginary time Green's function $G(\mathbf{q},\tau)$ and real-frequency spectral function $B(\mathbf{q},\omega)$

$$G(\mathbf{q},\tau) = \int_0^\infty \frac{d\omega}{\pi} \frac{e^{-\tau\omega} + e^{-(\beta-\tau)\omega}}{1 + e^{-\beta\omega}} B(\mathbf{q},\omega) \tag{4}$$

where $B(\mathbf{q},\omega) = S(\mathbf{q},\omega)(1 + e^{-\beta\omega})$.

In the practical calculation, we parametrize the $B(\mathbf{q},\omega)$ with a large number of equal-amplitude $\delta$-functions sampled at locations in a frequency continuum as $B(\omega) = \sum_{i=0}^{N_w-1} a_i\delta(\omega - \omega_i)$. Then the relationship between Green's function obtained from Eq. (4) and from QMC can be described by the goodness of fit $\chi^2$, i.e., $\chi^2 = \sum_{i=1}^{N_\tau}\sum_{j=1}^{N_\tau}(G_i - \overline{G}_i)C_{ij}^{-1}(G_j - \overline{G}_j)$, where $\overline{G}_i$ is the average of QMC measurement and $C_{ij}$ is covariance matrix $C_{ij} = \frac{1}{N_B(N_B-1)}\sum_{b=1}^{N_B}(G_i^b - \overline{G}_i)(G_j^b - \overline{G}_j)$, with $N_B$ the number of bins. Then we update the series of $\delta$-functions in a Metropolis process, from $(a_i, \omega_i)$ to $(a_i', \omega_i')$, to get a more probable configuration of $B(\mathbf{q},\omega)$. The weight for a given spectrum follows the Boltzmann distribution $P(B) \propto \exp(-\chi^2/2\Theta)$, with $\Theta$ a fictitious temperature chosen in an optimal way so as to give a statistically sound mean $\chi^2$-value, while still staying in the regime of significant fluctuations of the sampled spectra so that a smooth averaged spectral function is obtained. The resulting spectra will be collected as an ensemble average of the Metropolis process within the configurational space of $\{a_i, \omega_i\}$, as detailed in refs. [28,29,31,32].

## Data availability
The data that support the findings of this study are available from the corresponding author upon reasonable request.

## Code availability
All numerical codes in this paper are available upon request to the authors.

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

## Acknowledgements

We are indebted to Ziyu Chen, Wentao Jin, Andreas Weichselbaum, Yao Shen, and Yuesheng Li for stimulating discussions. This work was supported by the Ministry of Science and Technology of China through the National Key Research and Development Program (Grant Number 2016YFA0300502), the Strategic Priority Research Program of the Chinese Academy of Sciences (Grant Number XDB28000000), the National Science Foundation of China (Grant Numbers 11421092, 11574359, 11674370, 11974036, 11874115, and 11834014), and Research Grants Council of Hong Kong Special Administrative Region of China through 17303019 and the Aspen Center for Physics, which is supported by National Science Foundation grant PHY-1607611. B.-B.C. was supported by the German Research foundation, DFG WE4819/3-1. W.L. was supported by the Fundamental Research Funds for the Central Universities. We thank the Center for Quantum Simulation Sciences at Institute of Physics, Chinese Academy of Sciences, the Computational Initiative at the Faculty of Science at the University of Hong Kong, the Tianhe-1A platform at the National Supercomputer Center in Tianjin, and Tianhe-2 platform at the National Supercomputer Center in Guangzhou for their technical support and generous allocation of CPU time.

## Author contributions

H.L. and Y.D.L. contributed equally to this work. W.L., Y.Q., and Z.Y.M. initiated the work. H.L. and B.B.C. performed the calculations of thermodynamics. X.T.Z. and X.L.S. performed first-principle calculations of the band structure and Y.D.L. performed the dynamical property simulations. All authors contributed to the analysis of the results. Z.Y.M. and W.L. supervised the project.

## Competing interests

The authors declare no competing interests.
