## [Peer Review File · Nature Communications]

Reviewers' comments:

Reviewer #1 (Remarks to the Author):

Report on manuscript 220640 by Li et al:

This is a very interesting manuscript in which the authors propose a new interpretation of the properties of the compound TmMgGaO_4 and make several observations and predictions connected to the presence of a Kosterlitz-Thouless phase. First thought to be simply an example of a classical J_1 - J_2 model on the triangular lattice (as proposed in Ref. 25), this compound has been recently proven to have strong quantum fluctuations, as demonstrated by the inelastic neutron scattering data reported in Ref. 26. In Ref. 26, the authors also suggest that the system should be described by a J_1 - J_2 Ising model in a transverse field, and using a semiclassical spin-wave analysis, they manage to fit the neutron spectrum with a rather large value of the transverse field.

The present manuscript goes significantly beyond Ref. 26 in several respects:

- Using state-of-the-art numerical simulations, they propose a new fit of the experimental data based on a much smaller transverse field.
- They point out that this new set of parameters puts the system in the clock-ordered phase of the transverse field Ising model.
- They further point out that, in this phase, the system is expected to have a rich phase diagram as a function of temperature, the low-temperature clock phase being separated from the high temperature paramagnetic phase by a critical floating phase bounded by two Kosterlitz-Thouless transitions.
- They argue that available experimental data already show evidence in favour of this scenario, and they propose further experiments to extract the exponent η that would confirm their interpretation.
- They interpret the minimum around M as a roton minimum.

I think that the interpretation is sound and interesting. In particular, in the light of their interpretation, TmMgGaO_4 appears as a unique example of this floating phase in the context of rare earth magnetism. In my view, these results are definitely interesting enough to warrant publication in Nature Communications.

Now, since a lot has to do with the best parameters to describe the system, and with the most appropriate way to calculate the thermodynamics and the dynamics of the system, I think the authors should consider strengthening further their case by considering the following suggestions:

- The set of parameters proposed by the authors is used throughout, and the experimental data are well reproduced. So this is clearly a valid set of parameters. What is unclear however is to which extent this is the only one. Could the authors be more precise about how they came to this set, and which kind of error bars they can estimate from their analysis? In particular, how sensitive are the results to the parameter J_2 ? Would one still get a minimum at M without J_2 ? I am really curious to see how such a small value can be extracted from the experimental data.
- On page 7, the authors claim that the parameters proposed in Ref. 26 would put the system in the disordered magnetic phase, and they quote a critical value of the transverse field $\Delta_c = 0.8 J_1$. I am surprised by this value. Without J_2 , the critical value at zero temperature is $1.6 J_1$. Do the authors imply that a J_2 as small as $0.046 J_1$ reduces this critical value by a factor 2? This seems to contradict preliminary results I had obtained (and not published) three years ago, which would

rather put the boundary around 1.2. Note that this value would still agree with the argument of the authors that the parameters of Ref. 26 would put the system in the disordered phase. Note also that the results had been obtained on a 30×30 lattice with $\beta = 6$.

- In the same spirit, the authors claim on page 3 of the supplemental material that the critical value of J_2 to enter the stripe phase would be around 0.1 for $\Delta=0.54 J_1$. Again according to the preliminary results I had obtained three years ago, the critical value was more around 0.06 (still putting the system with the new set of parameters proposed by the present authors in the clock-ordered phase).

- While the analytical continuation is not fully controlled (and to a certain extent controversial), QMC can give access very accurately to the bottom of the excitation without resorting to analytical continuation. I think it would be useful to show such unbiased results (on top of the results obtained with analytical continuation) to strengthen the case.

Report written by Frédéric Mila.

Reviewer #2 (Remarks to the Author):

October 2019 Nature Comm.

This manuscript presents a detailed theoretical/numerical study of the physics underlying the rare-earth compound TmMgGaO_4 , a realization of the Ising triangular-lattice with an intrinsic transverse field. It is very rare to be able to realize this model in an actual material, but this is precisely what occurs in TmMgGaO_4 due to the combination of a peculiar crystal electric field environment for the non-Kramers Tm^{3+} ions.

Overall, this manuscript presents excellent research and although it deploys sophisticated numerical tools and presents subtle theoretical concepts, it reads extremely well. In fact, I am ready to say it is the best paper I've read on the subject of TmMgGaO_4 and it the paper feels like a classic.

In the paper, the authors resort to tensor network and quantum Monte-Carlo based techniques to calculate the thermo-magnetic and dynamical properties of TmMgGaO_4 and propose that this solid-state compound realizes a Kosterlitz-Thouless (KT) phase that resembles that of superfluid helium. The argument that TmMgGaO_4 hosts the celebrated KT phase is convincing based on predictions for the static and dynamical susceptibilities of a realistic model Hamiltonian. On that front, the authors construct a microscopic Hamiltonian that significantly advances and improves over previous studies. For instance, the authors claim to have "fixed the inadequate treatment of quantum fluctuations in the linear spin-wave approximation" in ref. 26, a claim that I have verified to be corrected through my own spin-wave calculations. A side result of the work is to establish a protocol of acquiring equilibrium and dynamic experiments of strongly correlated quantum material with intrinsic transverse field leading to non-trivial quantum tunneling phenomena. This is important because many such quantum magnets are becoming available to experimentalists.

Given the quality of the presentation and of the research, and the delay already incurred due to my late review (which I apologize for), I recommend this work be published in Nature Communications with the highest priority in its present form.

Reviewer #3 (Remarks to the Author):

The manuscript "The Ghost of Vanishing Stripe Order in the Triangular Quantum Ising Magnet TmMgGaO₄" by Han Li et al reports the quantum Monte Carlo (QMC) and Tensor Network (TN) studies of the triangular lattice Ising antiferromagnetic model in TmMgGaO₄. This model ideally predicts the celebrated Kosterlitz Thouless (KT) transition in the purely J₁ limit. The authors modeled the neutron scattering results in TmMgGaO₄ from Ref 26 (published in Nature Communications) which emphasized the effects of quantum fluctuations, and bulk characterization data from Ref 24-25. In particular, they have considered an alternative model with a relatively large splitting $\Delta = 0.54 J_1$, but a relatively smaller $J_2 = 0.05 J_1$ which preferentially favors the clock model over a stripe order (which onsets at $J_2 > 0.2 J_1$).

I find that the contents of the paper are well-grounded in the discussions or the references. The paper obviates the drawbacks of a purely classical approach of the Linear Spin Wave Theory and introduces the effects of quantum fluctuations via Quantum Monte Carlo, which represents an improvement over previous works. It represents an important and alternate viewpoint of the Hamiltonian in TMGO as compared to previous publications, which then makes a closer connection with the celebrated Kosterlitz Thouless Model. In order to test the conclusions, the paper has made provisions for future experiments that will provide validity to the hypotheses presented in the paper, such as gapped/gapless M point stripy modes, softening of the gap and broadening of the lineshapes. If the argument of proximity to KT turns out to be true then it is indeed interesting materials development which will raise hopes in the community.

Overall, I liked the presentation style and the conclusions in the paper. The thermodynamic fits to the data in Fig 2. look satisfactory. However, before I can make a judgment in regards to the publication of the manuscript in Nature Communications, I would like the authors to first address the following points:

(i) In Fig 1, "... shows the magnetic stripe order, with the red sites for spin up and blue ones spin down, on three sublattices A, B, and C. Vortex antivortex pair is created by flipping simultaneously two spins within the red oval in left subpanel". This visual connection to KT and the clock and stripy orders, as well as the language in the text and figure caption, is unclear both in the picture and in the text.

(ii) I don't think that the main argument of the paper is the presence of a "ghost peak" (which b.t.w. is a very weird and Halloweeny name). The authors should consider using a more scientific word such as a 'remnant' or 'residual'. Also along the same lines, I don't think that the remnant of a stripy phase can be such an important aspect that it finds its place in the title of the paper. The authors should re-write the title and the abstract to put more emphasis on the alternative model itself, which is interesting in its own right to make TMGO concur better with a KT model.

(iii) In page 7, what do the authors mean when they say that they 'adjust the x-axis to lay the model calculations' ? It is unclear. Did they scale, or did they add some constant term? What is the adjustment that is meaningful?

(iv) Also in Page 7, I was unable to figure how the authors can come to the conclusion that they have a 'quantitative' agreement: "The roton-like modes at the M-point present in the QMC-SAC results, with an energy gap of 0.4 meV, in quantitative agreement with that in Ref 26.". Did the authors fit to the neutron data intensity explicitly?

(v) Also, I was not able to understand what the authors mean by a 'gap'. All I can see in the data in ref 26 that M point is a saddle point, and not a bottom of the band. A 'gap' cannot be defined for a saddle point. There is no minima at the M-point in the spectrum. Do the authors suggest that their data shows a dip (with a well-defined gap) at the M-point?

As a result, I am also ambiguous about the 'roton' arguments in the discussion.

(vi) In page 8, "linewidth near the rotor like minima is substantially broadened, suggesting strong fluctuations and vortex proliferation in the system." do the authors have any direct proof (or a reference?) that the vortices will lead to a broadening?

Perhaps, can the authors make a prediction towards how much broadening will be quantitatively seen as a function of temperature, or vortex-antivortex pair concentration, considering that the M point excitations are somehow related to the vortex pair formation, something which can be checked with INS on a triple-axis instrument?

(vii) How are the authors sure whether the "ghost peaks" at the M-point are remnants of stripe-like fluctuations? Can the authors perform filtering of the spectrum, just keeping the M-point intensities, and checking whether they indeed arise from the stripy? In other words, I am asking the authors to critically examine what other varieties of orders can lead to an M-point static (or quasielastic low frequency) intensity, and provide grounds to preclude those.

(viii) On the same point as above, when the authors say: "move the vortex on the triangular lattice, like in a "tight-binding" model, by flipping spins on further neighboring sites, which naturally leads to a quadratic-type low-energy dispersion near point along the G-M-G." is there a calculation (or a reference) which suggests that the roton-like dip will be produced for such a motion? Can the authors show a spectrum in the proximate stripy phase (say by increasing J_2/J_1 by infinitesimal amounts) that the M point 'ghost' is stronger, and in the real space, this clearly leads to stripy?

Essentially I am not sure that the M-point intensity is continuously connected to the stripy phase as the authors propose, hence I am asking to test.

(ix) On a more general note, I find it surprising that the authors suggest a coexistence of two orders - the stripy and the vortex-antivortex phases - without any energy scale separation between them. Generally, when two different dynamic orders form, at any given energy this order occupies the entire Brillouin Zone. Two different phenomena can generally they occur at different energy scales (see for example Nature Materials 15, 733 (2016)), or involve different dimensions (See for example Nature Materials 4, 329 (2005)), or different places (i.e., via a (classical) domain separation). What exactly is happening here? Perhaps the authors can clarify the dynamic scattering and the energy scales of the vortex-antivortex formation, as opposed to those which arise from the stripy phase and the clock order in order to clarify.

Overall, I find that the paper presents some interesting conclusions of the nature of the intermediate temperature phase in TMGO and have outlined some future experiments which the community can perform to prove these hypotheses. However, at this point, it is not yet clear to me that these are the only unique conclusions from their simulations, or if the authors have performed the diligence to preclude other alternate scenarios. Hence, at this point, I reserve my recommendation for publication.

Manuscript NCOMMS-19-27650

Title: Kosterlitz-Thouless Melting of Magnetic Order in the Triangular Quantum Ising Material $TmMgGaO_4$

Dear Referees,

We thank you very much for the valuable and very insightful remarks and comments, which have helped us to further improve our manuscript. In the following we give a point-by-point response to the comments. The text of reviewers are cited in blue, with our subsequent response in normal format. Note that all bibliographic citations below make direct reference to the bibliography in our manuscript.

Best regards,

Han Li, Yuan Da Liao, Bin-Bin Chen, Xu-Tao Zeng, Xian-Lei Sheng, Yang Qi, Zi Yang Meng, and Wei Li

Response to the first Reviewer's report

Reviewer #1 : *This is a very interesting manuscript in which the authors propose a new interpretation of the properties of the compound $TmMgGaO_4$ and make several observations and predictions connected to the presence of a Kosterlitz-Thouless phase.*

First thought to be simply an example of a classical J_1 - J_2 model on the triangular lattice (as proposed in Ref. 25), this compound has been recently proven to have strong quantum fluctuations, as demonstrated by the inelastic neutron scattering data reported in Ref. 26. In Ref. 26, the authors also suggest that the system should be described by a J_1 - J_2 Ising model in a transverse field, and using a semiclassical spin-wave analysis, they manage to fit the neutron spectrum with a rather large value of the transverse field.

The present manuscript goes significantly beyond Ref. 26 in several respects:

- Using state-of-the-art numerical simulations, they propose a new fit of the experimental data based on a much smaller transverse field.

- They point out that this new set of parameters puts the system in the clock-ordered phase of the transverse field Ising model.

- They further point out that, in this phase, the system is expected to have a rich phase diagram as a function of temperature, the low-temperature clock phase being separated from the high temperature paramagnetic phase by a critical floating phase bounded by two Kosterlitz-Thouless transitions.

- They argue that available experimental data already show evidence in favour of this scenario, and they propose further experiments to extract the exponent η that would confirm their interpretation.

- They interpret the minimum around M as a roton minimum.

Reply: Thanks for this excellent summary of our points over previous works on the material TMGO. We hope our work serves as a textbook example of using precision quantum many-body techniques to

FIG. R1: **Parameter fittings.** (a) The specific heat $C_m(T)$ curves with various Δ/J_1 and fixed $J_2 = 0$, obtained by XTRG simulations on the $YC6 \times 9$ lattice. In panel (b) we show the susceptibility $\chi(T)$ with various $\Delta/J_1 = 0.4, 0.5, 0.7$ (at $h=1$ kOe), which all clearly failed to fit the experimental data well. (c) shows the $C_m(T)$ results with J_2/J_1 ranging from 0.01 to 0.09 and a fixed (optimal) $\Delta/J_1=0.54$, and (d) depicts the susceptibility data correspondingly.

study 2D frustrated magnets and remind the community the danger of blindly using mean-field type of analysis without taking the quantum fluctuations into proper consideration. As for now, our methodology indeed helps achieve an accurate modeling of the material TMGO and make predictions on the exotic KT thermodynamics that are experimentally accessible.

Reviewer #1 : *I think that the interpretation is sound and interesting. In particular, in the light of their interpretation, $TmMgGaO_4$ appears as a unique example of this floating phase in the context of rare earth magnetism. In my view, these results are definitely interesting enough to warrant publication in Nature Communications.*

Reply: We thank Prof. Frédéric Mila for the strong recommendation for publication.

Reviewer #1 : *Now, since a lot has to do with the best parameters to describe the system, and with the most appropriate way to calculate the thermodynamics and the dynamics of the system, I think the authors should consider strengthening further their case by considering the following suggestions:*

- *The set of parameters proposed by the authors is used throughout, and the experimental data are well reproduced. So this is clearly a valid set of parameters. What is unclear however is to which extent this is*

the only one. Could the authors be more precise about how they came to this set, and which kind of error bars they can estimate from their analysis? In particular, how sensitive are the results to the parameter J_2 ?

Reply: Thanks for the question. The workflow of parameter fittings is as follows: we scan the parameters (J_1, J_2, Δ) to fit the specific heat $C_m(T)$, magnetic entropy $S_m(T)$, as well as the susceptibility $\chi(T)$, and find their most probable values. Given that, we compute the magnetization curves (at different T) and magnetic entropy S_m at finite fields h , and compare directly to experimental data, so as to ensure that the parameter set is adequate and optimal to model the material.

To be concrete, we show in Fig. R1 a small part of our simulation data in the scanning. In Fig. R1(a), we start with $J_2 = 0$ and scan various Δ values, and it is found that C_m curves are quite sensitive, in terms of the peak height as well as the overall shape, to different Δ values. By tuning Δ (while keeping $J_2 = 0$), we find $\Delta/J_1 = 0.4$ and $J_1 = 1.1$ meV can produce results in agreement with experimental C_m curves. However, with this set of parameters (as well as other Δ values) we clearly miss the experimental susceptibility line, as plotted in Fig. R1(b). It therefore suggests that a finite J_2 should be involved in the fittings.

After some scanning in the parameter space, in Fig. R1(c) we find $\Delta/J_1 = 0.54$ with $J_2/J_1 = 0.05$ can well reproduce both the specific heat curve in Fig. R1(c) and the magnetic susceptibility data in (d). To show how sensitive the fittings are with respect to J_2 , we also provide in Figs. R1(c,d) the simulated data with $J_2 = 0.01$ to 0.09 , from which we see that J_2 considerably influences both $C_m(T)$ and $\chi(T)$ curves. Regarding the error bar of parameter J_2 , the most probably regime where J_2/J_1 resides is between 0.03 and 0.05 (while 0.05 is still more preferable), i.e., uncertainty 0.01 .

With this parameter set $J_1 = 0.99$ meV, $\Delta/J_1 = 0.54$, and $J_2/J_1 = 0.05$ (as well as $g_{\parallel} = 13.212$, see its discussions in Sec. C of Supplementary Information) we have computed the magnetization curves at two different temperatures as well as entropy S_m at finite magnetic fields, and compare them to experiments in Fig. 2 of the manuscript. Note there we push the XTRG calculations to YC9 cylinder (width $W = 9$), and find the fittings equally good.

Beyond equilibrium properties, we have also computed dynamical properties $\omega(k)$ of an $L = 36$ large-scale system. With the same parameter set, we see excellent agreement in Fig. R2(a). We also provide the $J_2 = 0$ data in Fig. R2(b), which do not fully agree with experimental results. Therefore, it can be clearly seen that the comparisons to dynamic spectrum also signifies the necessity of a finite J_2 .

All in all, these fittings as well as direct comparisons lead us unambiguously to the conclusion that the above parameters can precisely describe the material TMGO. We thank Prof. Mila again for this excellent suggestion to strengthen our point, and have now added a new Sec. B in the Supplementary Information that includes Fig. R1 to provide more of the fitting details.

Reviewer #1 : *Would one still get a minimum at M without J_2 ?*

Reply: Yes, as shown in Fig. R2(b). Remind that these are updated QMC-SAC results for $L = 36$ and we have also directly extracted the gap values from the imaginary time decay of the dynamical spin-spin

FIG. R2: **The dynamical spin spectrum with (a) $J_2/J_1 = 0.05$ and (b) $J_2 = 0$.** The contour plots show computed spectra obtained by QMC-SAC, and the white diamonds represent the bottom-part excitation results without resorting to analytical continuation, as suggested by the referee. Both numerical estimates of dispersions, SAC and bottom-part excitation, are in excellent agreement in (a) as well as in (b). The experimental INS results are shown as green dots, they are in a very good agreement with $J_2/J_1 = 0.05$ simulation in (a), while not so good with $J_2 = 0$ data in (b).

correlation functions in QMC, the roton modes are also there even without J_2 (while at a higher energy).

Reviewer #1 : *I am really curious to see how such a small value can be extracted from the experimental data.*

Reply: As shown in Figs. R1 and discussed above, it turns out that the fitting to thermodynamics is quite sensitive to J_2 , with which we can determine its precise value in TMGO. By scanning over a large number of parameters, we find that a finite (but small) $J_2/J_1 = 0.05$ is necessary to accurately reproduce the thermodynamic properties. Moreover, as mentioned above, the dynamical results in Fig. R2 also indicate the relevance of a finite J_2 .

Reviewer #1 : *- On page 7, the authors claim that the parameters proposed in Ref. 26 would put the system in the disordered magnetic phase, and they quote a critical value of the transverse field $\Delta_c = 0.8 J_1$. I am surprised by this value. Without J_2 , the critical value at zero temperature is $1.6 J_1$. Do the authors imply*

that a J_2 as small as $0.046 J_1$ reduces this critical value by a factor 2? This seems to contradict preliminary results I had obtained (and not published) three years ago, which would rather put the boundary around 1.2. Note that this value would still agree with the argument of the authors that the parameters of Ref. 26 would put the system in the disordered phase. Note also that the results had been obtained on a 30×30 lattice with $\beta = 6$.

Reply: We thank the careful referee and sorry that this is actually due to the confusion in convention. The specific Hamiltonian we are referring to is Eq. (1) in our manuscript, where spin-1/2 operators are used. Therefore, $\Delta_c/J_1 \sim 0.8$ we quoted in the main text corresponds to “ $1.6J_1$ ” in the Pauli-matrix convention, which Prof. Mila probably adopts. So there is no contradiction.

That is to say, we have assumed that a small $J_2/J_1 = 0.05$ will not affect strongly the critical value Δ_c between the clock and paramagnetic phases. We thank Prof. Mila for letting us know the more precise $\Delta_c/J_1 \simeq 0.6$ (in our convention), with which our conclusion still holds. Indeed, the Δ value should be below Δ_c since TMGO is found in a clock-order phase in experiments. In a follow-up project, we are exploring the phase diagram with parameters J_2 and Δ , etc, and will try to nail down these boundaries precisely. Up to now, QMC results on up to $L = 48$ torus confirm that our set of parameters in the manuscript stabilizes the clock order at low temperature.

Reviewer #1 : *- In the same spirit, the authors claim on page 3 of the supplemental material that the critical value of J_2 to enter the stripe phase would be around 0.1 for $\Delta=0.54 J_1$. Again according to the preliminary results I had obtained three years ago, the critical value was more around 0.06 (still putting the system with the new set of parameters proposed by the present authors in the clock-ordered phase).*

Reply: Also thanks for letting us know the more precise the critical J_2 value. Adding J_2 to the transverse-field triangular lattice Ising model can actually give rise to quite rich physics, which we are currently working on in a separate project.

Reviewer #1 : *- While the analytical continuation is not fully controlled (and to a certain extent controversial), QMC can give access very accurately to the bottom of the excitation without resorting to analytical continuation. I think it would be useful to show such unbiased results (on top of the results obtained with analytical continuation) to strengthen the case.*

Reply: Thanks, we have followed the suggestion to fit the exponential decay of dynamic spin-spin correlation function in the imaginary time, and in this way, obtain the bottom of the excitation without resorting to analytical continuation, as suggested by the referee. As shown in the Fig. R2 (as well as in the updated Fig. 4 of the revised manuscript), the comparison of such bottom is indeed consistent with the spectra obtained from SAC, in both cases with $J_2 = 0$ and $0.05J_1$.

Response to the second Reviewer's report

Reviewer #2 : *This manuscript presents a detailed theoretical/numerical study of the physics underlying the rare-earth compound $TmMgGaO_4$, a realization of the Ising triangular-lattice with an intrinsic transverse field. It is very rare to be able to realize this model in an actual material, but this is precisely what occurs in $TmMgGaO_4$ due to the combination of a peculiar crystal electric field environment for the non-Kramers Tm^{3+} ions.*

Overall, this manuscript presents excellent research and although it deploys sophisticated numerical tools and presents subtle theoretical concepts, it reads extremely well. In fact, I am ready to say it is the best paper I've read on the subject of $TmMgGaO_4$ and it the paper feels like a classic.

In the paper, the authors resort to tensor network and quantum Monte-Carlo based techniques to calculate the thermo-magnetic and dynamical properties of $TmMgGaO_4$ and propose that this solid-state compound realizes a Kosterlitz-Thouless (KT) phase that resembles that of superfluid helium. The argument that $TmMgGaO_4$ hosts the celebrated KT phase is convincing based on predictions for the static and dynamical susceptibilities of a realistic model Hamiltonian.

On that front, the authors construct a microscopic Hamiltonian that significantly advances and improves over previous studies. For instance, the authors claim to have "fixed the inadequate treatment of quantum fluctuations in the linear spin-wave approximation" in ref. 26, a claim that I have verified to be corrected through my own spin-wave calculations. A side result of the work is to establish a protocol of acquiring equilibrium and dynamic experiments of strongly correlated quantum material with intrinsic transverse field leading to non-trivial quantum tunneling phenomena. This is important because many such quantum magnets are becoming available to experimentalists.

Given the quality of the presentation and of the research, and the delay already incurred due to my late review (which I apologize for), I recommend this work be publish in Nature Communications with the highest priority in its present form.

Reply: We thank the second Reviewer for the highly positive assessment, and the very strong recommendation of our work. We truly believe that there are a lot more to explore in this intriguing material as well as other triangular rare-earth quantum magnets, both theoretically and experimentally.

Response to the third Reviewer's report

Reviewer #3 : *The manuscript “The Ghost of Vanishing Stripe Order in the Triangular Quantum Ising Magnet TmMgGaO₄” by Han Li et al reports the quantum Monte Carlo (QMC) and Tensor Network (TN) studies of the triangular lattice Ising antiferromagnetic model in TmMgGaO₄. This model ideally predicts the celebrated Kosterlitz Thouless (KT) transition in the purely J_1 limit. The authors modeled the neutron scattering results in TmMgGaO₄ from Ref 26 (published in Nature Communications) which emphasized the effects of quantum fluctuations, and bulk characterization data from Ref 24-25. In particular, they have considered an alternative model with a relatively large splitting $\Delta = 0.54 J_1$, but a relatively smaller $J_2 = 0.05 J_1$ which preferentially favors the clock model over a stripe order (which onsets at $J_2 > 0.2 J_1$).*

I find that the contents of the paper are well-grounded in the discussions or the references. The paper obviates the drawbacks of a purely classical approach of the Linear Spin Wave Theory and introduces the effects of quantum fluctuations via Quantum Monte Carlo, which represents an improvement over previous works. It represents an important and alternate viewpoint of the Hamiltonian in TMGO as compared to previous publications, which then makes a closer connection with the celebrated Kosterlitz Thouless Model. In order to test the conclusions, the paper has made provisions for future experiments that will provide validity to the hypotheses presented in the paper, such as gapped/gapless M point stripy modes, softening of the gap and broadening of the lineshapes. If the argument of proximity to KT turns out to be true then it is indeed interesting materials development which will raise hopes in the community.

Reply: We thank the referee for his/her high assessment of our work. Indeed, we believe that we have found out a set of parameters that can describe precisely the material TMGO. This discovery points to an interesting and open area in which the state-of-art quantum many-body simulation techniques can be used to explicitly explain the experimental results and predict further experiments. There are a lot more that can be explored in this material and along this line of research.

TMGO constitutes a very ideal platform for investigating exotic and rich quantum phases and effects in the transverse-field triangular lattice Ising (TLI) system. This includes the experimental detection of KT physics in TMGO, exploration of phase diagram of TLI under magnetic fields, and the effects of finite J_2 in excitations, etc. These questions call for further theoretical as well as experimental studies in this model system TMGO.

Reviewer #3 : *(i) In Fig 1, “... shows the magnetic stripe order, with the red sites for spin up and blue ones spin down, on three sublattices A, B, and C. Vortex antivortex pair is created by flipping simultaneously two spins within the red oven in left subpanel”. This visual connection to KT and the clock and stripy orders, as well as the language in the text and figure caption, is unclear both in the picture and in the text.*

Reply: In Fig. 1(c) of the manuscript, we have shown that the magnetic excitations, in terms of spin flipping on top of the stripy configuration, can be related to the vortex-antivortex pair through a pseudo-spin picture. To be concrete, the z -components of spin (m^z) are colored red and blue, for spin up and

spin down, respectively, on three sublattices (A, B, C) of the triangular lattice. m^z represents the on-site physical degrees of freedom. On the other hand, the (black) arrows plotted in the face of each triangle are emergent degree of freedom (pseudo spin) $\psi = m_A^z + e^{i2\pi/3} m_B^z + e^{i4\pi/3} m_C^z$, representing the U(1) order parameter.

In the left panel of Fig. 1(c) of the manuscript, we show the stripe order configuration (spin up on A sublattice and spin down on B and C sublattices), which can be mapped to a peculiar pseudo-spin pattern, though with zero vorticity. When one starts to flip a pair of spins (e.g., in the red oval), two topological “defects” with opposite charges are created, representing a vortex-antivortex pair in the pseudo-spin configurations. As one further moves the flipped spin around in the background of the stripe order, the pseudo-spin vortex also moves, with a dispersion $\epsilon(k)$. As plotted in Fig. R5, $\epsilon(k)$ exhibits a saddle point at momentum M (see explanations below). In this way, the stripe order and vortex-antivortex (roton) excitation are delicately related.

Furthermore, since the strip pattern is only short ranged in the intermediate temperature regime, the vortex and antivortex are bounded in the small stripy cluster until it reaches the upper KT transition, where the roton modes soften and vortices proliferate.

We thank the third Referee for this valuable comment. We have now critically examined related texts in the manuscript and have made some careful revisions to it, in order to improve their readability. Fig. 1(c) has also been revised, so as to be more self-evident.

Reviewer #3 : *(ii) I don't think that the main argument of the paper is the presence of a “ghost peak” (which b.t.w. is a very weird and Halloweeny name). The authors should consider using a more scientific word such as a ‘remnant’ or ‘residual’. Also along the same lines, I don't think that the remnant of a stripy phase can be such an important aspect that it finds its place in the title of the paper. The authors should re-write the title and the abstract to put more emphasis on the alternative model itself, which is interesting in its own right to make TMGO concur better with a KT model.*

Reply: We thank the third Referee for this great suggestion, and have now revised the title as “Kosterlitz-Thouless melting of magnetic order in the triangular quantum Ising material TmMgGaO₄” to put more emphasize on the KT physics in TMGO. Accordingly, the abstract and main text are also revised.

Reviewer #3 : *(iii) In page 7, what do the authors mean when they say that they ‘adjust the x-axis to lay the model calculations’ ? It is unclear. Did they scale, or did they add some constant term? What is the adjustment that is meaningful?*

Reply: Our theoretical curves are computed in the natural unit $J_1 = 1$, while the x-axis unit in experimental C_m curve is Kelvin. Therefore, “adjustment” here means finding the proper energy scale $J_1 = 0.99$ meV in the fittings. Thanks for asking and we have now rephrased there with a better wording “rescale”.

Reviewer #3 : *(iv) Also in Page 7, I was unable to figure how the authors can come to the conclusion that they have a ‘quantitative’ agreement: “The roton-like modes at the M-point present in the QMC-SAC*

results, with an energy gap of 0.4 meV, in quantitative agreement with that in Ref 26.”. Did the authors fit to the neutron data intensity explicitly?

Reply: As shown in the comparisons Fig. 4(a) of the revised manuscript, we see excellent agreement between QMC-SAC dispersion and the experimental curve. Furthermore, if we check the precise value, the roton gap estimated from experiment [Fig. 4(c) in Ref. [26]] is about 0.39 meV, which is in very good agreement with our numerical result. Allow us to emphasize, the parameters we used in the QMC-SAC simulations are not obtained from fitting the neutron data: they are obtained from fitting the thermal data including specific heat, susceptibility, and magnetization curves, etc, and the simulation agrees excellently with the neutron experiments by direct comparison.

Reviewer #3 : *(v) Also, I was not able to understand what the authors mean by a ‘gap’. All I can see in the data in ref 26 that M point is a saddle point, and not a bottom of the band. A ‘gap’ cannot be defined for a saddle point. There is no minima at the M-point in the spectrum. Do the authors suggest that their data shows a dip (with a well-defined gap) at the M-point? As a result, I am also ambiguous about the ‘roton’ arguments in the discussion.*

Reply: The M point indeed constitutes a saddle point. When cut along Γ - M - Γ line, a roton dip can be seen clearly, based on which we define the roton gap. In another word, the roton gap is the excitation energy gap exactly at the M saddle point.

Reviewer #3 : *(vi) In page 8, “linewidth near the roton like minima is substantially broadened, suggesting strong fluctuations and vortex proliferation in the system.” do the authors have any direct proof (or a reference?) that the vortices will lead to a broadening? Perhaps, can the authors make a prediction towards how much broadening will be quantitatively seen as a function of temperature, or vortex-antivortex pair concentration, considering that the M point excitations are somehow related to the vortex pair formation, something which can be checked with INS on a triple-axis instrument?*

Reply: Thanks for this nice comment and great suggestion. Matter of fact, in Ref. [26] (Fig. 4c) as well as Fig. 4 of our manuscript, it can be seen that the dispersion line near M point gets continuously broadened as T increases. We provide more direct numerical simulations in Fig. R3 (now also added in Fig. 4 of the manuscript), which shows how the intensity vs. ω curves change as T increases, at the fixed M point. This constitutes another interesting prediction that can be checked in a triple-axis spectrometer.

Regarding the reason for the line broadening, as we have proposed, the M roton in TMGO represents a bounded pair of vortex and antivortex, with a well-defined dispersion $\epsilon(k)$ at low T (as witnessed in Fig. 4 of our manuscript as well as in Fig. 4c of Ref. [26]). Once the two constituents of roton become unbounded (as T approaches the upper KT transition), and then the total momenta and energy of the “roton” are sum of two, i.e., $k = k_1 + k_2$ and $\epsilon_k = \epsilon_{k_1} + \epsilon_{k_2}$. k_i and ϵ_{k_i} are momentum and energy of i -th independent vortex ($i = 1, 2$), respectively. This would naturally lead to the broadening of dispersion line shape near M . Note this argument follows the line in Ref. [30], where the dispersion curve broadens due to spinon deconfinement.

FIG. R3: **The M point intensity vs. ω at various temperatures.** The calculations are obtained by QMC-SAC on $L = 36$ systems.

Reviewer #3 : (vii) *How are the authors sure whether the “ghost peaks” at the M-point are remnants of stripe-like fluctuations? Can the authors perform filtering of the spectrum, just keeping the M-point intensities, and checking whether they indeed arise from the stripy? In other words, I am asking the authors to critically examine what other varieties of orders can lead to an M-point static (or quasielastic low frequency) intensity, and provide grounds to preclude those.*

Reply: This is a great question. The classical spin orders in $J_1 - J_2$ TLI have been thoroughly investigated in Ref. [38]. We have plotted them in Fig. R4, along with their static structure factor $S(\mathbf{q})$. One can see clearly that only orders in Figs. R4(c,d) have peaks at M point. Fig. R4(d) corresponds to the stripy order that agrees with our simulated structure factor data (in large J_2 case), while the one in (c) has a Γ peak that is absent in our results. Therefore, the only classical Ising configuration that corresponds to our structure factor data is Figs. R4(d), i.e., stripy order.

Furthermore, we can also argue semiclassically that large $J_2 > 0$ favors a stripy order: In the presence of J_2 , it leads to an energy estimate of $-J_1 - J_2$, while the up-up-down order [A,B sublattice spin up and C sublattice down, see Fig. R4(b)] to $-J_1 + 3J_2$, and the order in Figs. R4(c) to 0, etc. Therefore, it is clear that the stripy configuration is energetically favorable in large $J_2 (> 0)$ limit. Moreover, in a recent study of another triangular lattice magnet AgNiO_2 with large single-axis anisotropy, the existence of a magnetic stripe order was revealed, which was ascribed to the relatively large J_2 in that compound [39].

We can also argue it the other way around. The Bragg peak at $M = (1/2, 1/2)$ corresponds to a π phase shift, i.e., antiferromagnetic configuration, along \mathbf{a} and \mathbf{b} (see primitive vector in Fig. 1 of the manuscript), while the correlation is ferromagnetic along $\mathbf{a} + \mathbf{b}$ and $\mathbf{a} - \mathbf{b}$. This clearly corresponds to a stripy phase, in the context of TLI model.

Overall, thanks for the question and we have now added Fig. R4 in the Supplementary Information Sec. E.

Reviewer #3 : (viii) *On the same point as above, when the authors say: “move the vortex on the triangu-*

FIG. R4: **Classical orders of the $J_1 - J_2$ TLI model and their static structure factor $S(\mathbf{q})$.** Note the $S(\mathbf{q})$ data are computed in a 12×12 cluster, symmetrized, and normalized by the number of site in it.

lar lattice, like in a “tight-binding” model, by flipping spins on further neighboring sites, which naturally leads to a quadratic-type low-energy dispersion near point along the G-M-G.” is there a calculation (or a reference) which suggests that the roton-like dip will be produced for such a motion?

Reply: We thank the Reviewer for raising this important question. Indeed, we should provide more details for this argument, which actually follows a perturbative calculation of the transverse-field Ising model.

When restricted in a subspace of configurations with only one pair of spins flipped (while others remain in the classical stripy order, in the small Δ limit), we consider a “tight-binding” model

$$H = \epsilon_0 + \sum_i \sum_{\delta} t(c_i^\dagger c_{i+\delta} + \text{h.c.}) \quad (\text{R1})$$

on the triangular lattice, where i labels the lattice site, and $|i\rangle$ labels a state with a spin flipped at site i . $\delta = \mathbf{a}, \mathbf{b}, \mathbf{a} - \mathbf{b}$ denotes nearest neighboring sites (\mathbf{a}, \mathbf{b} are primitive vectors shown in Fig. 1 of the

FIG. R5: “Tight-binding” dispersion on triangular lattice and saddle point at M . (a) shows the contour plot of dispersion $\epsilon(k)$, and (b,c) show the cuts along paths 1 and 2, respectively.

manuscript). Remember that a second-order process related to S^x terms in the Hamiltonian can actually tunnel between $|i\rangle$ and $|j\rangle$, so $t \sim \Delta^2$ (in the small Δ limit), given i and j are a pair of nearest neighboring sites.

For the sake of simplicity, we set $\epsilon_0 = 0, t = 1$, and take Fourier transformation of Eq. R1. The resulting dispersion $\epsilon(k) = 2[\cos(k \cdot \mathbf{a}) + \cos(k \cdot \mathbf{b}) + \cos(k \cdot (\mathbf{a} - \mathbf{b}))]$ is plotted in Fig. R5(a). By cutting along the Γ - M - Γ path, we observe a quadratic low-energy dispersion near the minimum as shown in Fig. R5(b). On the other hand, as shown in Fig. R5(c) the M point constitutes a maximal along K - M - K path. Therefore, it is evident that the M point indeed constitutes a saddle point in the dispersion. We have now added Fig. R5 and related discussions in Supplementary Information Sec. G.

Reviewer #3 : *Can the authors show a spectrum in the proximate stripy phase (say by increasing J_2/J_1 by infinitesimal amounts) that the M point ‘ghost’ is stronger, and in the real space, this clearly leads to stripy? Essentially I am not sure that the M -point intensity is continuously connected to the stripy phase as the authors propose, hence I am asking to test.*

Reply: We show our structure factor data in Fig. R6 attached below (and meanwhile updated in Fig. S4 of Supplementary Information). A scenario of continuously enhanced stripy correlation as J_2/J_1 increases can be seen clearly in Figs. R6(b-g). This can be more clearly visualized by plotting the M point strength vs. J_2/J_1 in Fig. R6(a), we observe S_M increases continuously as J_2 increases, and the system enters the stripy phase when $J_2 > J_{2c}$.

Note the critical J_{2c} to break the clock order is small ($J_{2c} \sim 0.1$ as estimated by in Fig. R6, and ~ 0.06 by Reviewer 1 Frédéric Mila), therefore the two orders strongly compete. We thus believe the remnant M -point peak at intermediate T represent instability towards the stripy order.

Reviewer #3 : *(ix) On a more general note, I find it surprising that the authors suggest a coexistence of two orders - the stripy and the vortex-antivortex phases - without any energy scale separation between*

FIG. R6: **Static structure factors $S(\mathbf{q})$ at various couplings J_2/J_1 .** In (a) we collect the M -point intensity and plot it vs. J_2/J_1 at four different temperatures, from which we see clearly that S_M is continuously connected to that in the stripy phase. The contour plots of $S(q)$ are shown in panels (b-g), with various J_2/J_1 values (0 to 0.2). We fix $\Delta/J_1=0.54$, $J_1=0.99$ meV, and $T=2.24$ K.

them. Generally, when two different dynamic orders form, at any given energy this order occupies the entire Brillouin Zone. Two different phenomena can generally they occur at different energy scales (see for example Nature Materials 15, 733 (2016)), or involve different dimensions (See for example Nature Materials 4, 329 (2005)), or different places (i.e., via a (classical) domain separation). What exactly is happening here? Perhaps the authors can clarify the dynamic scattering and the energy scales of the vortex-antivortex formation, as opposed to those which arise from the stripy phase and the clock order in order to clarify.

Reply: We agree, indeed, that two phenomena in a given system generally happens at different energy/length/temperature scales.

Matter of fact, here in TMGO we see two temperature scales, T_l and T_h . Below the lower temperature scale T_l (lower KT transition), there exists a clock order, while in the intermediate temperature $T_l < T < T_h$ there exist a floating KT phase with vortex-antivortex excitations. Above T_h , paramagnetism sets in.

In the KT phase, the clock order already melts, and the “ghost” peak at M point signals the instability towards a stripy order. The latter does not develop a true long-range order in the intermediate T regime, and remains only short-ranged. To summarize, the two phenomena (clock order and KT phase) are controlled and separated by temperature scales. Anyway, we thank the Reviewer for pointing us two interesting references which we enjoy reading and find them deserving a place in our reference list.

Reviewer #3 : *Overall, I find that the paper presents some interesting conclusions of the nature of the intermediate temperature phase in TMGO and have outlined some future experiments which the community*

can perform to prove these hypotheses. However, at this point, it is not yet clear to me that these are the only unique conclusions from their simulations, or if the authors have performed the diligence to preclude other alternate scenarios. Hence, at this point, I reserve my recommendation for publication.

Reply: Thanks for the positive assessment and interest of our work. As elaborated above, within the transverse-field Ising model, we can concretely preclude other scenarios. In addition, we think our correct modeling of TMGO is just a beginning and it definitely calls for more experimental as well as theoretical studies on this intriguing quantum material and other similar triangular magnets.

All in all, we sincerely hope our response and additional data above, as well as revisions to the manuscript and Supplementary Information, can convince Reviewer 3 that our work lies on a solid ground and merits the publication in the prestigious Nature Communications.

Summary of changes made

The summary of the major changes in the resubmitted text are shown below. We also corrected further minor typos, and the major changes include:

1. We have revised the title as “Kosterlitz-Thouless Melting of Magnetic Order in the Triangular Quantum Ising Material TmMgGaO_4 ” to emphasize the KT physics in TMGO. The abstract is also modified accordingly.
2. We replace fitted parameter “ $g_J = 1.101$ ” with the more conventional (yet equivalent) notation “ $g_{\parallel} = 13.212$ ” in the manuscript, i.e., in Page 6 as well as the caption of Fig. 2.
3. We add three new references, i.e., Refs. [38], [48], and [49], and in the updated bibliography.
4. In Fig. 1(c), we add in the bottom part the tracks of the winding process and mark the “cores” of “vortex” and “anti-vortex” with ± 1 in order to reach a better explanation. The caption is also revised accordingly.
5. In Fig. 4, we have updated the QMC dynamic results with a larger size $L = 36$ simulation and corrected typos in temperatures T due to a misunderstanding in convention. As suggested by the reviewers, we now show both the spectra obtained from SAC and directly fitting the imaginary time decay of the dynamic spin-spin correlation functions. A very good consistency is seen in Fig. 4(a). Also we add the M point intensity vs. ω at various temperatures in Fig. 4(d) to show the continuously broadened linewidth near the rotonlike minima, as suggested by the third Reviewer.
6. We add three new sections in Supplementary Information, i.e., “TLI parameter fittings”, “classical spin orders and their static structure factors”, “saddle point in the triangular tight-binding model”, as new Secs. B, E, and G, respectively.

7. Fig. S5 in Supplementary Information (Sec. D) is updated, with additional static structure factor data of various J_2/J_1 parameters.

REVIEWERS' COMMENTS:

Reviewer #1 (Remarks to the Author):

I think the authors have done a great job at answering all referees' questions, and at modifying the manuscript accordingly.

Regarding my questions about the boundaries of the phase diagram, the authors correctly realized that I had been working with Pauli matrices, explaining the discrepancy between their results and mine.

This is a very interesting piece of work, and I strongly recommend its publication in Nature Communications in its present form.

Reviewer #3 (Remarks to the Author):

Upon re-examining the revised manuscript with the revised title "Kosterlitz-Thouless Melting of Magnetic Order in the Triangular Quantum Ising Material TmMgGaO_4 " by Wei Li et al, I am happy to see that the authors have made the attempt to address all the points and in most cases they have done it satisfactorily.

The only point where I think that the manuscript should be further improved and the text can be clarified is in connection with the discussion on the saddle point and the roton spectrum where I invite the authors to refer to historical papers on this point, and also in conjunction to the recent interesting manuscripts on this topic : R Verresen et al., Nature Physics volume 15, pages750–753(2019), and Ferrari & Becca, PHYSICAL REVIEW X 9, 031026 (2019), perhaps discussing the softening of the M point spectrum and the 'melting' in the light of the above findings.

Beyond that, I am convinced that the manuscript is a very valuable addition to the literature of triangular lattice antiferromagnets which are deeply studied in search for spin liquid excitations and Kosterlitz Thouless physics, proposing that TMGO is a strong contender for the later. I am convinced that the manuscript will incite substantial interest in the community to experimntally find some of the predictions made by the authors such as in new INS proposals. Once the author addresses the above point, I will be willing to accept the manuscript for publication in Nature Communications

Manuscript NCOMMS-19-27650

Title: Kosterlitz-Thouless Melting of Magnetic Order in the Triangular Quantum Ising Material TmMgGaO₄

Response to the first Reviewer's report

Reviewer #1 : *I think the authors have done a great job at answering all referees' questions, and at modifying the manuscript accordingly.*

Regarding my questions about the boundaries of the phase diagram, the authors correctly realized that I had been working with Pauli matrices, explaining the discrepancy between their results and mine.

This is a very interesting piece of work, and I strongly recommend its publication in Nature Communications in its present form.

Reply: Thanks for the appreciation of our reply and revision, and for the strong recommendation of our work.

Response to the third Reviewer's report

Reviewer #3 : *Upon re-examining the revised manuscript with the revised title "Kosterlitz-Thouless Melting of Magnetic Order in the Triangular Quantum Ising Material TmMgGaO₄" by Wei Li et al, I am happy to see that the authors have made the attempt to address all the points and in most cases they have done it satisfactorily.*

Reply: Thanks for agreeing on our reply, and for your valuable comments which we have taken as incentives to further examine and strengthen our points.

Reviewer #3 : *The only point where I think that the manuscript should be further improved and the text can be clarified is in connection with the discussion on the saddle point and the roton spectrum where I invite the authors to refer to historical papers on this point, and also in conjunction to the recent interesting manuscripts on this topic: R Verresen et al., Nature Physics volume 15, pages 750-753(2019), and Ferrari and Becca, PHYSICAL REVIEW X 9, 031026 (2019), perhaps discussing the softening of the M point spectrum and the 'melting' in the light of the above findings.*

Reply: Thanks for raising this interesting point which we would very much like to elaborate further. Rotonlike modes has been predicted theoretically to exist in triangular lattice Heisenberg (TLH) antiferromagnets [45–48] (note that all citations in this reply make direct reference to the bibliography in our main paper) and later confirmed experimentally [13, 14], whose nature is still under ongoing research [49, 50]. Notably, it has been proposed that the M rotons in TLH can be softened by further enhancing spin frustration

(and thus quantum fluctuations) [50] or thermal fluctuations [51], which melts the long-range or incipient semi-classical 120° order, driving the system into “intermediate” liquid-like spin states.

So, there indeed bears great similarity in the scenarios between TLH and here TMGO in this work, through thermal/quantum softening of rotons. Although the exact nature of intermediate phase in TLH remains illusive, here in TMGO we determine the intermediate-temperature regime as floating KT phase and thus significantly promote the understandings of rotons in 2D triangular quantum magnets.

Reviewer #3 : *Beyond that, I am convinced that the manuscript is a very valuable addition to the literature of triangular lattice antiferromagnets which are deeply studied in search for spin liquid excitations and Kosterlitz Thouless physics, proposing that TMGO is a strong contender for the later. I am convinced that the manuscript will incite substantial interest in the community to experimentally find some of the predictions made by the authors such as in new INS proposals. Once the author addresses the above point, I will be willing to accept the manuscript for publication in Nature Communications.*

Reply: Thanks for the recommendation, indeed, and please see our response to your question above.